# Bone Cements Used for Hip Prosthesis Fixation: The Influence of the Handling Procedures on Functional Properties Observed during In Vitro Study

**DOI:** 10.3390/ma15092967

**Published:** 2022-04-19

**Authors:** Alina Robu, Robert Ciocoiu, Aurora Antoniac, Iulian Antoniac, Anca Daniela Raiciu, Horatiu Dura, Norin Forna, Mihai Bogdan Cristea, Ioana Dana Carstoc

**Affiliations:** 1Faculty of Material Science and Engineering, University Politehnica of Bucharest, 313 Splaiul Independentei, District 6, 060042 Bucharest, Romania; ciocoiurobert@gmail.com (R.C.); antoniac.aurora@gmail.com (A.A.); antoniac.iulian@gmail.com (I.A.); 2Academy of Romanian Scientists, 54 Splaiul Independentei, District 5, 050094 Bucharest, Romania; 3S.C. HOFIGAL EXPORT IMPORT S.A, 2 Intrarea Serelor Str., District 4, 042124 Bucharest, Romania; daniela_raiciu@yahoo.com; 4Faculty of Medicine, University Lucian Blaga of Sibiu, 2A Lucian Blaga Str., 550169 Sibiu, Romania; ioana.carstoc12@gmail.com; 5Department of Orthopedics and Traumatology, University of Medicine and Pharmacy Gr.T.Popa Iasi, 16 University Str., 700115 Iasi, Romania; norin.forna@umfiasi.ro; 6Department of Morphological Sciences, Carol Davila University of Medicine and Pharmacy, 37 Dionisie Lupu Str., District 2, 020021 Bucharest, Romania; bogdan.cristea@umfcd.ro

**Keywords:** bone cements, biomaterials, mechanical properties, orthopedic, handling

## Abstract

The failure of hip prostheses is a problem that requires further investigation and analysis. Although total hip replacement is an extremely successful operation, the number of revision surgeries needed after this procedure is expected to continue to increase due to issues with both bone cement types and cementation techniques (depending on the producer). To conduct a comparative analysis, as a surgeon prepared the bone cement and introduced it in the body, this study’s team of researchers prepared three types of commercial bone cements with the samples mixed and placed them in specimens, following the timeline of the surgery. In order to evaluate the factors that influenced the chemical composition and structure of each bone cement sample under specific intraoperative conditions, analyses of the handling properties, mechanical properties, structure, and composition were carried out. The results show that poor handling can impede prosthesis–cement interface efficacy over time. Therefore, it is recommended that manual mixing be avoided as much as possible, as the manual preparation of the cement can sometimes lead to structural unevenness.

## 1. Introduction

Biomaterials are now used successfully not just in dentistry [1,2] and orthopedic surgery [3,4,5] but also in many medical specializations such as neurosurgery [6], ophthalmology [7], gynecology [8,9], cardiovascular surgery [10,11], general surgery [12,13,14,15] and maxillofacial surgery [16].

Bone cements are defined as mixtures of substances consisting of a powder phase and a liquid phase that, after mixing and homogenization, form a paste that can harden and self-stabilize once implanted in the body [17,18,19,20]. These materials have the ability to be modeled to ensure the fixation of the implant and act as an interface between bones and the implant material. Bone cements are widely used as materials for endoprosthesis replacement, vertebroplasty and cranioplasty. The two main types are calcium phosphate cements (CPCs) and polymethylmethacrylate bone cements (PMMAs) [21]. The selection of bone cements is carried out according to clinical needs.

For a cement to properly perform the clinical function for which it was created, it must simultaneously possess several characteristics: the ability to properly transmit static and dynamic loads, biocompatibility, properties similar to those of bone (e.g., elasticity), high fatigue resistance, crack resistance, resistance to abrasive processes, wear resistance, high coefficient of friction, relatively short link creation time, adequate polymerization temperature, high vibration damping factor, ease of handling, in vivo hardening in perfect time, nontoxicity, close to neutral pH during attachment, a contraction as small as possible during stabilization, good radiopacity and porosity. Although the requirements are known, the perfect cement has not been found so far; each of the materials used to date has its limitations. In current surgical interventions, the most frequently used cements can be classified as acrylic bone cements and calcium phosphate bone cements [22,23,24].

The mechanical resistance of the total hip prosthesis and particularly the adhesion quality between the implant and the bone primarily depend on the nature of the cement used and its mechanical and geometrical characteristics [25]. Acrylic bone cements are part of a group of materials that are created directly in their environment. They are obtained from PMMA, a simple liquid polymer/monomer system that hardens in the cold. The combination of the two components results in a viscous mass with a stabilization time of 7–10 min, until mechanical stability is reached. It is currently the most widely used biomaterial for fixing prostheses in arthroplasty and has a high performance due to its very good properties [26,27,28,29,30,31]. The properties and applications of acrylic bone cement are different, as shown in Figure 1 [32].

The general disadvantages of acrylic bone cement are the lack of osteoconductivity; aseptic weakening over time [33]; exothermic reaction during polymerization with possible local necrosis; reduced mechanical properties; lack of bioactivity [34]; hypersensitivity to the cement components [35]; possible cardiovascular reactions to acrylic bone cement [36]; possible leaching of the unreacted monomer into the surrounding tissues, leading to chemical necrosis [36]; and osteolysis due to the wear and tear of particles and debris from the bone cement.

Calcium phosphate cements (CPCs) are more similar to bone due to the presence of calcium phosphate [37,38]. 

Various papers describe the impact of cementation techniques on the clinical results of total hip or knee prosthesis, revealing that the cement mantle thickness influences the potential prosthesis migration and inflammatory reactions due to various wear particles [39,40,41,42]. Additionally, some failure analysis studies highlight the importance of intraoperative activities such as bone cement preparation [43,44,45,46].

The main function of bone cement is to adhere the prosthesis to the bone, resulting in the transference of body weight and mechanical loads from the prosthesis to the bone and the immediate immobilization of the prosthesis. This ensures the adequate fixation of the femoral component for from 10 to 15 years, but after this time failure is almost inevitable, even using the latest intraoperative methods for performing surgery and obtaining and handling cement. There are many reasons for endoprosthesis failure, such as the body’s excessive inflammatory response to infection, poor handling, soft tissue failure [47], aseptic loosening [48,49,50], improper mixing, failure to handle properly, poor stabilization, exothermic reaction [51] and the mechanical failure of the cement layer.

Since cementing is affected by various factors including lavage, hemostasis, bone density, the type of cement, mixing time, cement viscosity, timing, ambient temperature, cementing technique, component design and the surgeon’s experience, it is important to utilize reproducible cementing techniques [52,53]. There are different methods for handling the cement used to fix the femoral component, other methods for fixing the acetabular component and a completely different method for handling the cement used for BHR-type prostheses. As specified in the instructions for the use of the BIRMINGHAM HIP Resurfacing prosthesis (Surgical Technique), the low-viscosity cement is mixed and poured into the head implant. Alternatively, it can be drawn up into a bladder syringe and injected into the femoral component [54]. 

Although the stabilizing properties are primarily associated with the composition of the cement, it has been found that many other parameters affect the properties of bone cements, an important factor being the ambient temperature during handling. Higher temperatures decrease both the working time and the stabilization time, which can influence the handling of the cement [55]. The curing process is divided into four stages: (a) mixing, (b) waiting, (c) working and (d) hardening. The mixing can be carried out by hand or with the aid of centrifugation or vacuum technologies [18].

Previous results obtained by our group regarding the failure analysis of a hip-cemented prosthesis reveal that the improper preparation of the acrylic bone cements is one reason for prosthesis failure [56,57,58]. Figure 2 shows a hip resurfacing failure in which the defects due to the poor handling of the bone cement are highlighted.

Following these studies, the aim of this paper is to perform a comparative analysis of three different acrylic bone cements used in clinical practice for the same hip prosthesis fixation, Aminofix 1, Aminofix 3 and Simplex P, evaluating their surface and mechanical properties. Table 1 presents the stabilization parameters of some commercial acrylic bone cements.

## 2. Materials and Methods

### 2.1. Materials

#### Bone Cements Used

Bone cement samples were obtained directly in the operating room during surgery by mixing the liquid monomer with the powdered polymer. Cement was converted from a liquid to a solid state by an exothermic reaction. 

Table 2 presents the chemical compositions of the commercial products used in this research: Aminofix 1 (Groupe Lépine, Genay, France) encoded as sample 1, Aminofix 3 (Groupe Lépine, France) encoded as sample 2 and Simplex P (Stryker, Kalamazoo, MI, USA) encoded as sample 3.

The investigated bone cements are used for different procedures in orthopedic surgery. The difference between preparing the cement in the laboratory and preparing it in the operatory is that some surgical procedures take longer than others, so the cement cannot settle too quickly or too slowly. Therefore, some manufacturers produce several different types of bone cement, each designed for particular circumstances. 

Bone cement performance is directly linked to various parameters such as the mixing method, chemicals used, viscosity, porosity, antibiotics used in the cement composition, sterilization, working temperature, physical and mechanical properties and biocompatibility [61,62]. Additionally, different procedures for intraoperative handling of the bone cements are used in clinical practice. For classical hip arthroplasty, the bone cements are pressured inside the prepared bone and prosthesis components are inserted after into the cement (Figure 3), but in some specific cases such as hip resurfacing prostheses (e.g., Birmingham Hip Resurfacing Prosthesis), the bone cements are pressured inside the femoral head before the insertion of the prosthesis component into the prepared bone (Figure 4).

### 2.2. Methods

#### 2.2.1. Scanning Electron Microscopy Coupled with Energy Dispersive Spectroscopy

The morphology of the experimental samples and the elementary chemical composition were evaluated using a QUANTA INSPECT F scanning electron microscope (FEI Company, Eindhoven, The Netherlands) equipped with an energy-dispersive X-ray spectrometer detector (EDAX) (FEI Company, Eindhoven, The Netherlands) with a 132 eV resolution at MnK. 

#### 2.2.2. Contact Angle Measurements

The samples’ wettability was assessed by contact angle measurements performed using the KRÜSS DSA30 Drop Shape Analysis System.

The contact angle measurements were performed using the sessile drop method, with each measurement being repeated five times. The samples were fixed on a support to ensure flatness, and, using the automatic dosing system, drops of distilled water with variable volumes were deposited (5–15 µL) depending on the available flat surface. Once the drop of distilled water was deposited, an image was captured using the built-in measuring system camera. The obtained images were processed with the help of the ImageJ software, with which the contact angle was determined.

#### 2.2.3. Compressive Strength Measurements

The mechanical properties of the experimental samples were also evaluated, following their compression strength. The contact area of the acrylic cement with the bone is intensely mechanically stressed, so the determination of the compressive strength is very important [64,65]. 

This study was designed to determine the properties of the bone cements, mainly their structural characteristics, based on the viscosity information reported by the producer and their mechanical properties using a compression test. For the compression tests, the experiment was randomized. The sample size was influenced by the volume of available materials.

The compression tests were performed conforming to the ASTM D695 *Standard Test Method for Compressive Properties of Rigid Plastics* specifications using cylindrical specimens with a length double the diameter (ø20 mm × 40 mm). These were subjected to a mechanical compression test using a Walter + Bai LFV 300 device. Figure 5 schematically shows the system used for the compression test and the behavior of the cement during this test.

The compression test parameters were set according to the ASTM D695 specifications: The test was controlled by displacement using a speed of 1.3 ± 0.3 mm/min until the yield point was reached; then, the speed was increased to 5 mm/min. The repeatability, reproducibility and accuracy of the test results were verified against previous data [2,66,67]. The calibration and accuracy check of the testing machine were performed by the producer, and interlaboratory tests were performed regularly. Following the compression test, the software automatically generated the stress–strain diagram for all tested samples.

## 3. Results and Discussion

### 3.1. Scanning Electron Microscopy Coupled with Energy-Dispersive Spectroscopy Determinations

SEM analysis was carried out to evaluate the structure of the experimental samples and reveal the presence of potential agglomerations that could affect its homogeneity. The elemental composition of the samples was determined using EDS analysis.

Figure 6 shows the SEM images and corresponding EDS spectra for all the investigated samples. The SEM images highlight typical microstructures for PMMA-based cements: beads from the polymer powder; the matrix of the polymerized monomer; and the radiopacifying element—in this case, barium sulphate (BaSO4) and pores. 

The most homogeneous structure was observed for sample 3. For sample 1, SEM images display a slight tendency of barium sulphate to form agglomerates, and some pores could be observed. For sample 2, in contrast, the SEM images highlight agglomerations of the radiopacifying element due to an improper mixing of the cement’s components. As a first conclusion, we recommend centrifugal mixing for this type of cement and avoiding the manual mixing of the cement. 

The EDS analysis confirmed the composition of the commercial bone cement and highlighted the presence of the C and O from the two polymer phases (majority elements) and the Ba and S from the radiopacifying element composition.

Regarding the clinical significance of the results shown in this section, we should mention that the mixing is a very important step in bone cement preparation. If the bone cement is not well mixed, agglomerations can appear in the structure with a strong influence on the mechanical properties.

### 3.2. Contact Angle Determination

The values of the contact angles and the corresponding images for each investigated sample are presented in Table 3 and Figure 7.

The hydrophilic versus hydrophobic state of a material gives information about its biocompatibility because the surface properties strongly influence the interfaces between biomaterials and tissues [68]. In the literature, polymethylmethacrylate has an intrinsic contact angle of less than 90. The contact angles for the investigated samples in this study were very close to each other and to the PMMA angle, which was around 69° [69,70]. The obtained results indicated that the sample surfaces were hydrophilic. A hydrophilic surface reflects a good wettability and adhesiveness and thus a better osteointegration. Osteointegration is very important from the clinical point of view. An optimal contact angle for cell adhesion is around 55° [71,72], and we obtained the closest angle in this research with sample 3 (55.1°), followed by sample 1 and sample 2.

### 3.3. Compression Strength Determination

The aspects of the samples before and after performing the compression test are presented in Figure 8. For each investigated bone cement, we used three specimens (except for sample 1, for which 4 specimens were used), each of which had a cylindrical shape.

Following the compression test, the software automatically generates the stress–strain diagram for all tested samples as well as a comparative diagram with the values obtained from all samples (Figure 9). From this figure, we can observe, after the compression test, ductile behavior in all the investigated samples. The numerical values after the compression test performed on the investigated samples are presented in Table 4.

From the values obtained for yield strength, we observed that all the investigated bone cements fulfilled the minimum value established by the ASTM F451 standard (70 MPa). The best value was recorded for sample 3, an average of 92.94 MPa, followed by sample 1 (78.01 MPa) and sample 2 (77.55 MPa). Regarding the modulus of elasticity of the investigated samples, the values were close. The average modulus of elasticity of the Aminofix 1 sample (sample 1) was 2433.75 ± 263.68 MPa. The Aminofix 3 (sample 2) had an average modulus of elasticity of 2265.47 ± 582.60 MPa, and for the Simplex P (sample 3), the average was 2308.03 ± 32.15. Maximum stress represents the highest stress recorded during the test, and the two types of cement manufactured by the same manufacturer had similar values (~85 MPa).

In Figure 10, we compare the modulus of elasticity, the yield strength and the maximum stress variations, which were later tested using an ANOVA.

In a three-level ANOVA (for sample 1, we only used the first three values) of the modulus of elasticity and of the yield strength of the bone cement, the null hypothesis was accepted at α = 0.05; the expectation was that there would be no variation, as observed in the results shown in Table 5 and Table 6. 

Regarding the maximum stress, the results presented in Table 7 show that there was variation within the results; thus, the alternative hypothesis was accepted.

## 4. Conclusions

In the early stage of arthroplasty, it was difficult for orthopedic surgeons to identify the importance of the cementing technique and the bone cement selected, facts that led to numerous implant failures [73,74,75,76,77,78]. It is imperative for the clinical personnel who intraoperatively prepare the bone cements to precisely know all the required steps, mixing techniques, polymerization characteristics and handling procedures of the bone cements used in order to obtain a durable and high-quality fixation of the prosthesis components [55].

The investigated samples showed a structure that is typical for acrylic bone cements, with a tendency to form agglomerates due to improper mixing, specifically in the case of sample 2, which clearly demonstrated that manual product preparation can sometimes lead to structural unevenness. The wettability measured by determining the contact angle indicated the hydrophilicity of the investigated samples. In terms of mechanical properties, all investigated samples showed optimal values in accordance with the ASTM F451 standard; sample 3 achieved the best yield strength.

From the obtained results, we can conclude that the cement handling process (mixing stage, sticky/waiting stage, working stage) has an important role to play in the structural integrity and mechanical properties of bone cements and that manual mixing is to be avoided in the specific case of low-viscosity bone cements.

Although this study presents some limitations related to the small number of samples used and the reproducibility of the *in vivo* working conditions, we consider that the important information obtained will be of use to specialists in bone cement production as well as the clinicians who use bone cements.

In this paper, we found that there is no universal bone cement that can be used for all hip prostheses and that the selection and use of bone cements for hip prosthesis fixation must be correlated with hip prosthesis design, the fixation technique and devices used and the material properties.

## Figures and Tables

**Figure 1 materials-15-02967-f001:**
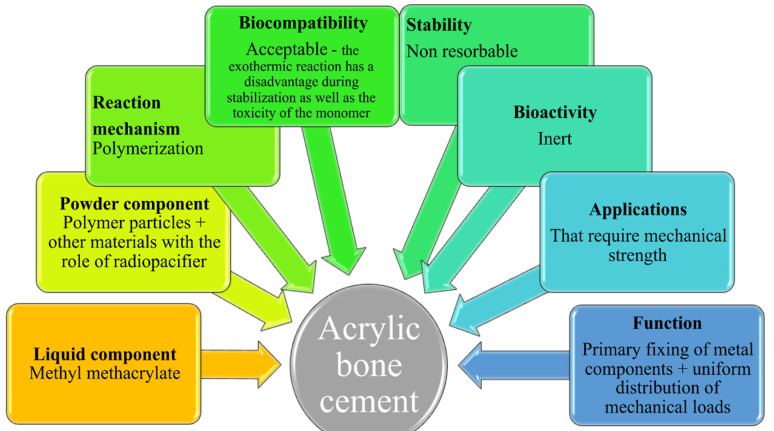
Schematic data of the acrylic bone cements.

**Figure 2 materials-15-02967-f002:**
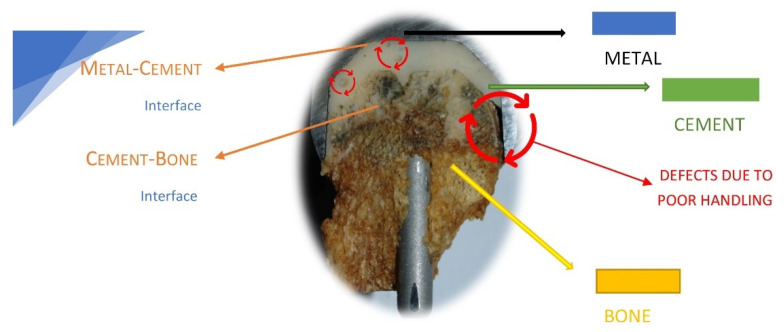
Schematic presentation of implant–cement and cement–bone interfaces.

**Figure 3 materials-15-02967-f003:**
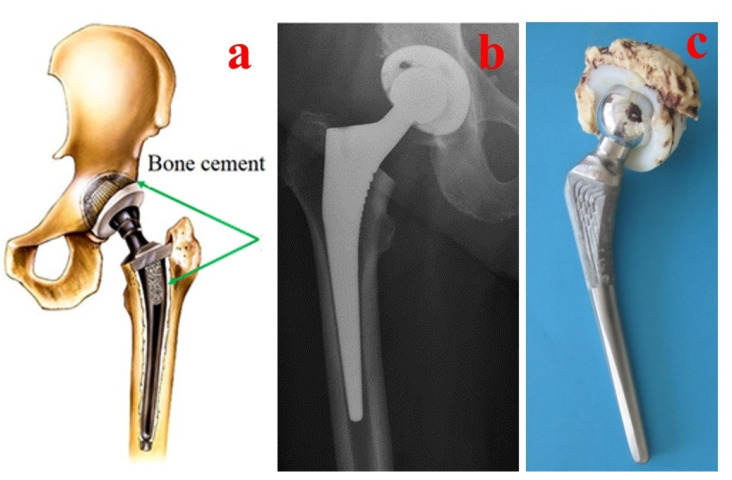
Aspects regarding the handling of bone cements for classical hip prosthesis fixation: (**a**) Bone cement application. (**b**) Postoperative radiography. (**c**) Failed hip prosthesis.

**Figure 4 materials-15-02967-f004:**
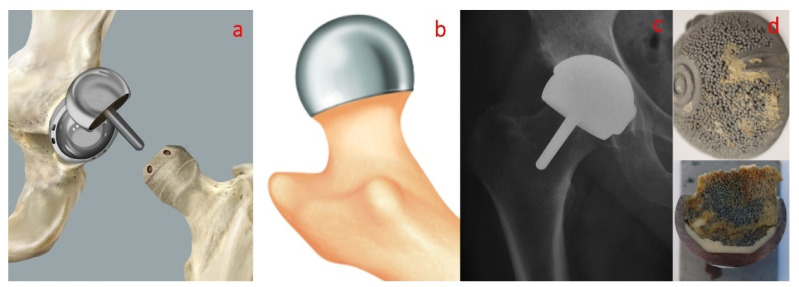
Aspects regarding the handling of bone cements for resurfacing hip prosthesis (e.g., BHR) fixation: (**a**) bone preparation; (**b**) cement pressure into the prosthetic components; (**c**) postoperative radiography; (**d**) failed BHR hip prostheses [63].

**Figure 5 materials-15-02967-f005:**
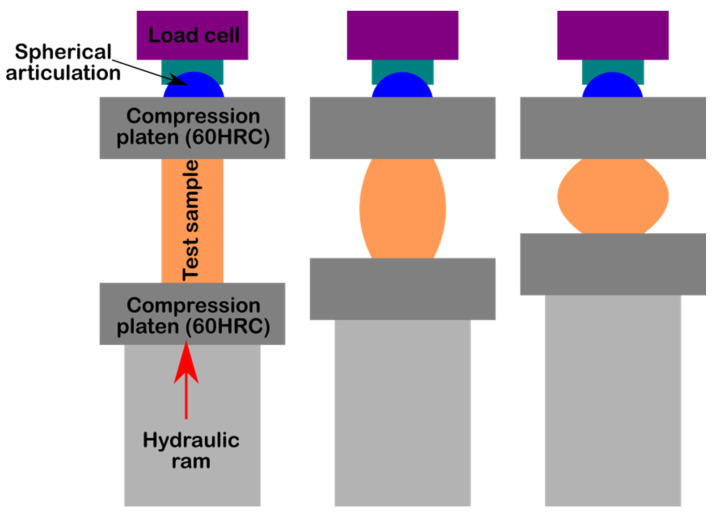
Schematic depicting the experimental set-up used for compression testing and sample behavior during the test (image not to scale).

**Figure 6 materials-15-02967-f006:**
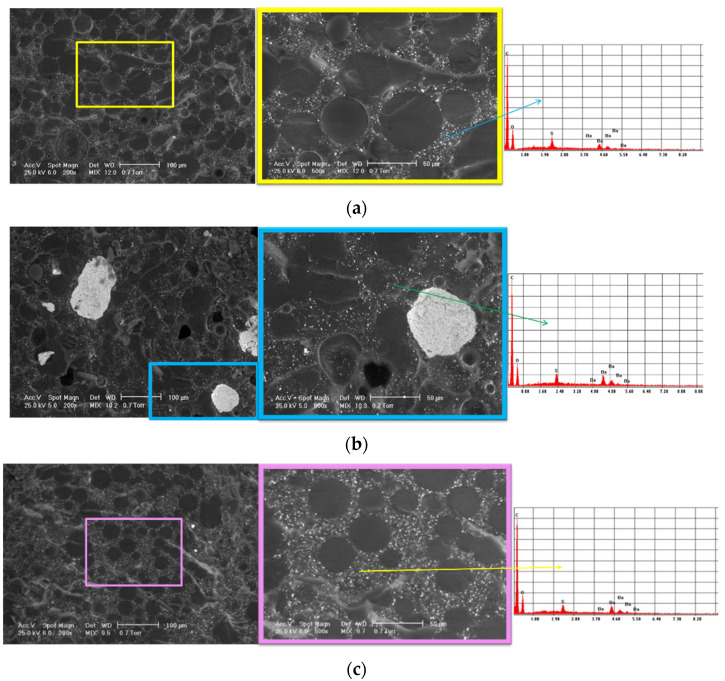
Representative SEM images and corresponding EDS spectra of the investigated samples: (**a**) sample 1; (**b**) sample 2; (**c**) sample 3.

**Figure 7 materials-15-02967-f007:**
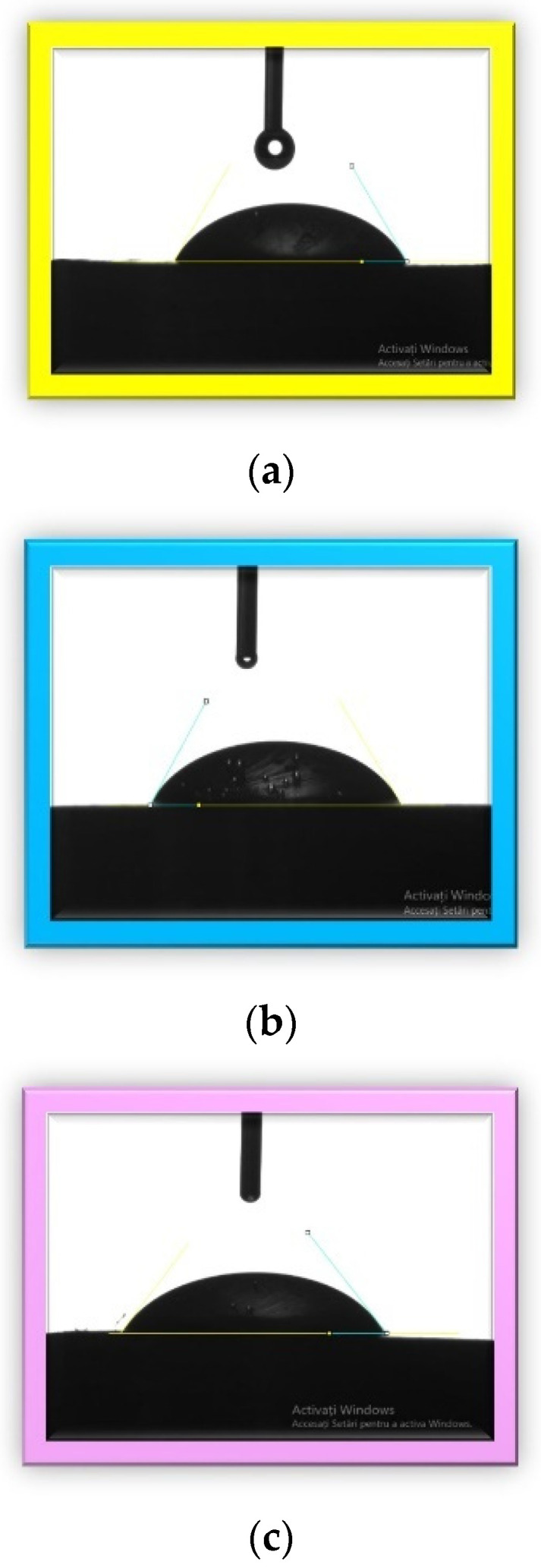
Images of liquid droplets on the surface of materials: (**a**) sample 1, (**b**) sample 2, (**c**) sample 3.

**Figure 8 materials-15-02967-f008:**
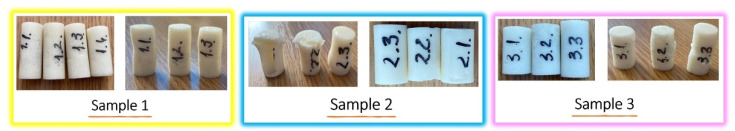
Images of the specimens before and after the compressive strength test (before compression—left; after compression—right.

**Figure 9 materials-15-02967-f009:**
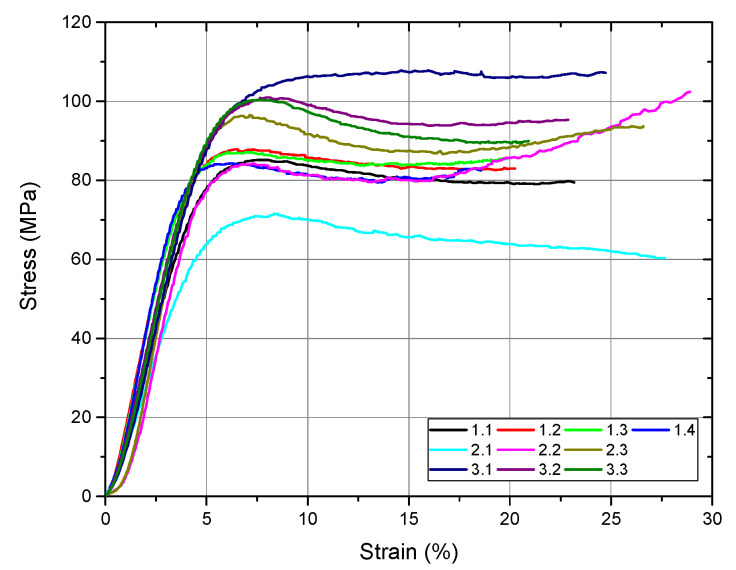
Diagram with the results for compression testing for all experimental samples (sample 1—code 1.1, 1.2, 1.3, 1.4; Sample 2—code 2.1, 2.2, 2.3; and Sample 3 code 3.1, 3.2, 3.3).

**Figure 10 materials-15-02967-f010:**
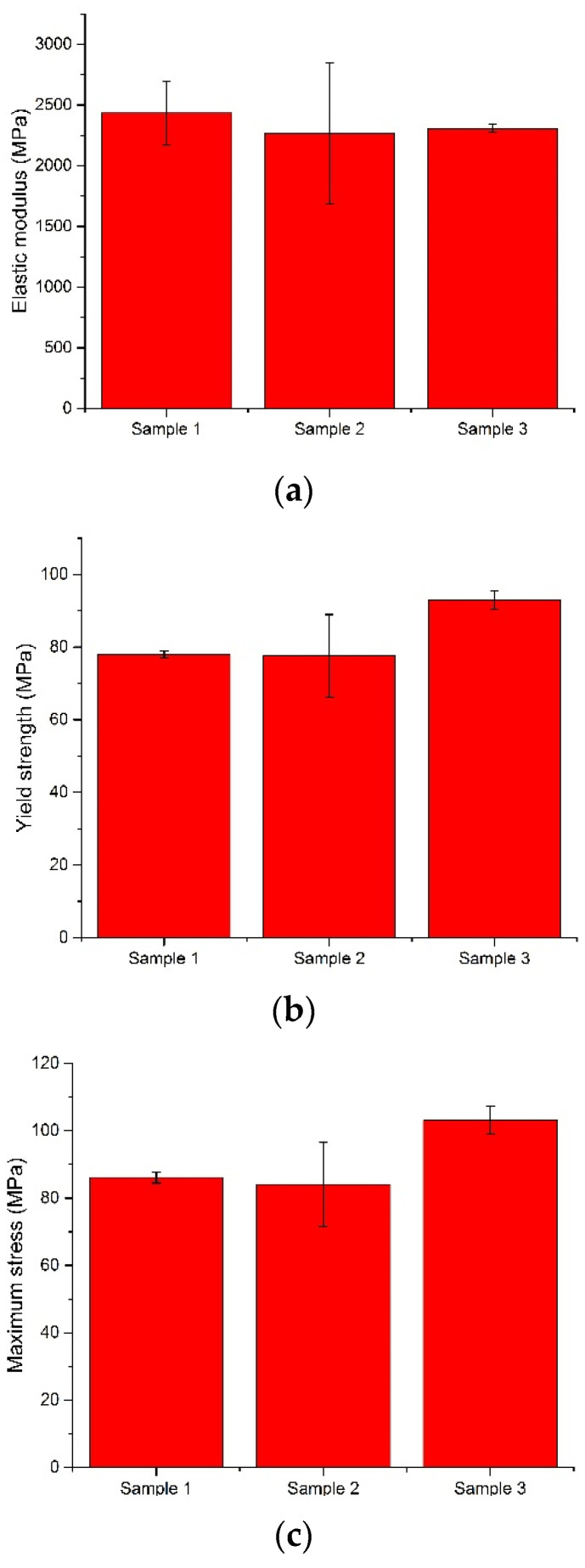
Comparison of the elastic modulus (**a**), the yield strength (**b**) and the maximum stress (**c**) using an ANOVA test.

**Table 1 materials-15-02967-t001:** Stabilization parameters of some commercial acrylic bone cements [59,60].

Bone Cement	Working Time (min)	Setting Time (min)	Peak Temperature (°C)
Fix 1 (Groupe Lepine)	3	7	57
Fix 2 (Groupe Lepine)	4	9	57
Aminofix 1 (Groupe Lepine)	4	8	57
Aminofix 3 (Groupe Lepine)	5	10	58
Simplex P(Stryker)	7	14.3	90
Palacos^®^ + G(Heraeus)	5	12.5	73
CMW1(DePuy CMW)	6.5	11	88
Osteopal V (Heraeus)	8	14	56
KyphX HV-R	8	20	56

**Table 2 materials-15-02967-t002:** Chemical composition of the experimental samples of acrylic bone cements.

		Sample 1	Sample 2	Sample 3
**Liquid component**		***14.4* g**	***16.4* g**	***18.79* g**
Methyl methacrylate	*Monomer*	*12.28*	*13.99*	*18.33*
Butyl methacrylate	*Monomer*	*1.90*	*2.16*	*-*
N, *N*-dimethyl-p-toluidine	*Activator*	*0.22*	*0.25*	*0.46*
Hydroquinone *	*Inhibitor*	*20* ppm	*20* ppm	*60* ppm
**Powder component**		***40* g**	***40* g**	***41* g**
Polymethyl methacrylate	*Pre-polymerized polymer*	*33.68*	*33.52*	*6.00*
Benzoyl peroxide	*Initiator*	*0.96*	*1.12*	*0.50*
Methyl methacrylate—styrene copolymer	*Pre-polymerized copolymer*	*-*	*-*	*30.00*
Barium sulphate	*Radiopaque agent*	*3.84*	*3.84*	*4*
Gentamicin sulfate	*Antibiotic*	*1.52*	*1.52*	*-*
Erythromycin	*Antibiotic*	-	-	*0.50*
Colistin Sulphomethate Sodium EP	*Antibiotic*	-	-	*3.00* million I.U.
Viscosity		standard	low	low

* Hydroquinone was added in relation to the liquid component.

**Table 3 materials-15-02967-t003:** Contact angle values for the experimental samples.

Sample/Bone Cement	Liquid	Contact Angle (*°*)
Sample 1	water	59.33 ± 2.68
Sample 2	water	60.67 ± 4.59
Sample 3	water	55.10 ± 3.57

**Table 4 materials-15-02967-t004:** The compression test results for the bone cement samples.

Sample Code	Modulus of Elasticity (MPa)	Yield Strength (MPa)	Maximum Stress (MPa)
1.1	2070.74	78.46	85.21
1.2	2456.87	77.44	87.88
1.3	2507.49	79.17	87.16
1.4	2699.90	76.98	84.30
Sample 1 average	**2433.75**	**78.01**	**86.14**
2.1	1610.59	65.59	71.60
2.2	2459.59	78.73	84.02
2.3	2726.22	88.33	96.55
Sample 2 average	**2265.47**	**77.55**	**84.06**
3.1	2272.86	94.59	107.84
3.2	2335.92	90.01	101.04
3.3	2315.32	94.22	100.41
Sample 3 average	**2308.03**	**92.94**	**103.10**

**Table 5 materials-15-02967-t005:** The ANOVA results for the elastic modulus.

Source for Variation	Sum of Squares	Degrees of Freedom	Mean Squares	F_0_	F_crit_
Treatments	6868.58	2	3434.29	0.0249	5.1439
Error	829,084	6	138,181		
Total	835,952	8			

**Table 6 materials-15-02967-t006:** The ANOVA results for the yield strength.

Source for Variation	Sum of Squares	Degrees of Freedom	Mean Squares	F_0_	F_crit_
Treatments	450.18	2	225.09	4.909	5.143
Error	275.10	6	45.85		
Total	725.28	8			

**Table 7 materials-15-02967-t007:** The ANOVA results for the maximum stress.

Source for Variation	Sum of Squares	Degrees of Freedom	Mean Squares	F_0_	F_crit_
Treatments	636.99	2	318.49	5.48	5.14
Error	349.02	6	58.17		
Total	986.01	8			

## Data Availability

Not applicable.

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
