# Peer review of "Bone Cements Used for Hip Prosthesis Fixation: The Influence of the Handling Procedures on Functional Properties Observed during In Vitro Study"

_materials, 2022, doi:10.3390/ma15092967_

Round 1

Reviewer 1 Report

1-The abstract is concise. The manuscript features the following chapters: 1. Introduction, 2. Materials and Methods, 3.Results and discussion, 4. Conclusions. Authors need to carefully revise the manuscript. Some suggestions are recommended.

2- Some figures and tables are not presented in the written form in the manuscript.

3-In the Introduction section, it is suggested to cite and discuss more, relevant recent literature related with cement materials and their behavior. Nevertheless, and between other, the authors can read recommended works about cement material and their characteristics as the following:

Bone metastatic tumor minimization due to thermal cementoplasty effect, clinical and computational methodologies. International Journal of Medical Engineering and Informatics, 2021, 13(1):35-43. DOI:10.1504/IJMEI.2020.10031214

4-Figures need to have higher resolution.

5- A figure representative of the entire experimental setup used for the compressive tests need be included.

6-In my opinion, to better explain the experimental setup, the authors should clarify whether there are any limitations during the experimental procedure.

7-Please give information about instrumentation calibration / accuracy / test cell velocity…for compressive tests. And about the experiments, was the repeatability / consistency verified?

8-Conclusions need to be augmented and future developments included.

Author Response

Dear reviewer,

            The Authors of the manuscript entitled “Bone cements used for hip prosthesis fixation: influence of the handling procedures on their functional properties” submitted to MATERIALS thank the reviewer for reviewing our manuscript. We are deeply grateful for the observations and comments which we addressed and feel that greatly increased the quality of our manuscript. Please find below the answers to all comments and suggestions.

  1. The abstract is concise. The manuscript features the following chapters: 1. Introduction, 2. Materials and Methods, 3. Results and discussion, 4. Conclusions. Authors need to carefully revise the manuscript. Some suggestions are recommended.

Answer:  Thank you for taking the time to review our manuscript and for your suggestions.

  1. Some figures and tables are not presented in the written form in the manuscript.

Answer: Thank you!  This observation is correct, and we have changed. The figures and tables were organized according to the suggestions.

  1. In the Introduction section, it is suggested to cite and discuss more, relevant recent literature related with cement materials and their behavior. Nevertheless, and between other, the authors can read recommended works about cement material and their characteristics as the following: Bone metastatic tumor minimization due to thermal cementoplasty effect, clinical and computational methodologies. International Journal of Medical Engineering and Informatics, 2021, 13(1):35-43. DOI:10.1504/IJMEI.2020.10031214

Answer: Thank you. We have revised the text to address your concerns and hope that it is now clearer. The suggested quote has been added to page 2 - position 51, along with other quotes.

  1. Figures need to have higher resolution.

Answer: We have revised the figures and we replaced them.  

  1. A figure representative of the entire experimental setup used for the compressive tests need be included.

Answer: A representative figure of the entire experimental (figure 5 in the revised manuscript) has been added.

  1. In my opinion, to better explain the experimental setup, the authors should clarify whether there are any limitations during the experimental procedure.

Answer: 6. The requested information has been added at page 6 (part 2.2.3.) of the revised manuscript. Although according to standard specifications a minimum of 5 specimens are to be tested, the number was reduced to 3 or 4 specimens, since the volume of available material per bone cement kit would suffice only for this number.

  1. Please give information about instrumentation calibration / accuracy / test cell velocity…for compressive tests. And about the experiments, was the repeatability / consistency verified?

Answer: The requested additions have been added at page 6 (part 2.2.3.) of the revised manuscript.

  1. Conclusions need to be augmented and future developments included.

Answer: Thank you for your suggestion. The Conclusions part were updated.

We would like to thank you to the referee for the valuable suggestions.

Reviewer 2 Report

Dear authors,

Thank you very much for your paper. In this paper, the authors presented a study entitled “ Bone cements used for hip prosthesis fixation: influence of the handling procedures on their functional properties” aiming to

to perform a comparative analysis of the different acrylic bone cements used in clinical practice for the same hip prosthesis fixation, respectively Aminofix 1, Aminofix 3 and Simplex P, following their surface and mechanical properties.

In general, the manuscript is very interesting and well-written. However, major corrections are required to improve the overall quality. An English-language review is required.

Moreover, there are several errors in the methodology(lack of statistical analysis) and discussion that need to be fixed before the publication of the manuscript:

My recommendations are the following:

Please insert on the title and abstract the type of the study in order to be immediately understandable for the reader. 

The abstract section is not well performed. The introduction section should be improved. Please consider in  this section the general disadvantages of bone cements.

Line 69: “Various papers describe the impact of cementation techniques on the clinical results of total hip or knee prosthesis, revealing that the cement mantle thickness influence the potential prosthesis migration and inflammatory reactions due to various wear particles [34].”  Various paper or one paper?

The materials and methods section needs to be totally reorganized with subheadings: Please insert study design, sample size calculation, Statistical analysis (Statystical analysis is missing).

However, it must be specified in detail how  sample size was calculated and the references. 

Discussion

  1. The discussion section should be improved by adding information regarding the clinical significance of the results obtained.
  2. Add some limitations of the study (In a separate paragraph)

Author Response

Dear reviewer,

            The Authors of the manuscript entitled “Bone cements used for hip prosthesis fixation: influence of the handling procedures on their functional properties” submitted to MATERIALS thank the reviewer for reviewing our manuscript. We are deeply grateful for the observations and comments which we addressed and feel that greatly increased the quality of our manuscript. Please find below the answers to all comments and suggestions.

  1. Please insert on the title and abstract the type of the study in order to be immediately understandable for the reader.

Answer:  We have changed the title and the abstract as suggested.

  1. The abstract section is not well performed. The introduction section should be improved.

Answer: We thank the reviewer for pointing this out. We have revised the abstract section.

“Failure of hip prostheses is a problem that requires further investigation and analysis. Although total hip replacement has been an extremely successful operation, the number of revision surgeries needed after such a procedure is expected to continue to increase due to issues with both bone cement types and cementation techniques (depending on producer). To make a comparative analysis, as a surgeon was preparing the bone cement and introducing it in the body, the team of researchers was preparing three types of commercial bone cements with the samples mixed and placed in specimens, following the timeline of the surgery. In order to evaluate the factors influencing the chemical composition and structure of each bone cement sample under specific intraoperative conditions, analyses of the handling properties and mechanical properties – structure and composition were carried out. The results show that poor handling can impede prosthesis cement interface efficacy over time. Therefore, it is recommended to avoid manual mixing as much as possible, as manual preparation of the cement can sometimes lead to structural unevenness.”

  1. Please consider in this section the general disadvantages of bone cements.

Answer: Thank you for your suggestion. We added the general disadvantages of bone cements in the introduction section.

“The general disadvantages of acrylic bone cement are: the lack of osteoconductivity; aseptic weakening happening over time [33]; exothermic reaction during polymerization with possible local necrosis; reduced mechanical properties; lack of bioactivity [34]; hypersensibility to the cement components [35]; possible cardiovascular reactions to acrylic bone cement [36]; possible leach of the unreacted monomer into the sur-rounding tissues leading to chemical necrosis [36]; osteolysis due to wear and tear of particles and debris from the bone cement.”

  1. Line 69: “Various papers describe the impact of cementation techniques on the clinical results of total hip or knee prosthesis, revealing that the cement mantle thickness influence the potential prosthesis migration and inflammatory reactions due to various wear particles [34].” Various paper or one paper?

Answer: Thank you for your suggestion. This observation is correct. We have completed with more papers.  

  1. The materials and methods section needs to be totally reorganized with subheadings:

Answer: We reorganized with subheadings the materials and methods section.

  1. Please insert study design, sample size calculation, Statistical analysis (Statystical analysis is missing).

Answer: The requested information has been added at page 6 with a schematic depicting the experimental set-up used for compression testing and sample behavior during the test, followed by the explication that was missing. Also, the statistical analysis was performed using ANOVA test (page 9 and page 10 in revised manuscript).

  1. However, it must be specified in detail how sample size was calculated and the references.

Answer: Thank you for pointing this out. Although according to standard specifications a minimum of 5 specimens are to be tested, the number was reduced to 3 or 4 specimens, since the volume of available material per bone cement kit would suffice only for this number.

  1. Discussion

The discussion section should be improved by adding information regarding the clinical significance of the results obtained.

Add some limitations of the study (In a separate paragraph)

Answer: Thank you for your suggestion. The discussions sections were updated according to your suggestion. Also, the section CONCLUSION was revised (page 11 in revised manuscript).

We would like to thank the referee again for taking the time to review our manuscript

Round 2

Reviewer 2 Report

The authors have adequately addressed the concerns of this reviewer.